# Hydroxytyrosol: A Promising Therapeutic Agent for Mitigating Inflammation and Apoptosis

**DOI:** 10.3390/pharmaceutics16121504

**Published:** 2024-11-22

**Authors:** Wafa Ali Batarfi, Mohd Heikal Mohd Yunus, Adila A. Hamid, Yi Ting Lee, Manira Maarof

**Affiliations:** 1Department of Physiology, Faculty of Medicine, Universiti Kebangsaan Malaysia, Jalan Yaacob Latiff, Bandar Tun Razak, Kuala Lumpur 56000, Malaysia; w.batarfi@hu.edu.ye (W.A.B.); adilahamid@ppukm.ukm.edu.my (A.A.H.); yitingyyl428@gmail.com (Y.T.L.); 2Department of Basic Medical Sciences, Hadhramout University College of Medicine, Al-Mukalla, Yemen; 3Department of Tissue Engineering and Regenerative Medicine, Faculty of Medicine, Universiti Kebangsaan Malaysia, Jalan Yaacob Latiff, Bandar Tun Razak, Kuala Lumpur 56000, Malaysia; manira@ppukm.ukm.edu.my

**Keywords:** hydroxytyrosol, inflammation, apoptosis, natural products

## Abstract

Inflammation and apoptosis are interrelated biological processes that have a significant impact on the advancement and growth of certain chronic diseases, such as cardiovascular problems, neurological conditions, and osteoarthritis. Recent research has emphasized that focusing on these mechanisms could result in novel therapeutic approaches that aim to decrease the severity of diseases and enhance patient outcomes. Hydroxytyrosol (HT), which is well-known for its ability to prevent oxidation, has been identified as a possible candidate for regulating both inflammation and apoptosis. In this review, we will highlight the multifaceted benefits of HT as a therapeutic agent in mitigating inflammation, apoptosis, and associated conditions. This review provides a comprehensive overview of the latest in vitro and in vivo research on the anti-inflammatory and antiapoptotic effects of HT and the mechanisms by which it works. Based on these studies, it is strongly advised to use HT as a bioactive ingredient in pharmaceutical products intended for mitigating inflammation, as well as those with apoptosis applications.

## 1. Introduction

Inflammation and apoptosis have a crucial role in the development of various chronic diseases, such as cardiovascular disease, neurological disorders, and metabolic syndromes [1,2,3,4]. The chronic activation of inflammatory pathways and disrupted apoptosis in cell regulation contribute to damage in tissues, the progression of diseases, and poor clinical outcomes [5,6]. As a result, there is an increasing interest in finding natural therapeutic compounds that can efficiently control these processes [7,8].

Hydroxytyrosol (HT), a phenolic molecule mostly present in olive oil, has attracted considerable interest due to its powerful antioxidant, ROS scavenger, and anti-inflammatory characteristics [9,10,11]. HT, an essential element of the Mediterranean diet, has been associated with numerous health benefits, such as decreased chances of cardiovascular diseases and enhanced neuroprotection [12,13,14]. Recent studies have determined that HT has effective anti-inflammatory and antiapoptotic effects [15,16,17,18,19,20]. Granados-Principal, et al. [21] stated that HT is a powerful antioxidant with potent anti-inflammatory properties, along with anti-platelet and anti-atherogenic effects. Another study conducted by Camargo, et al. [22] proved that consuming phenol-enriched olive oils in large quantities can suppress the activity of many genes associated with inflammation and atherosclerosis, leading to a reduced inflammatory response in peripheral blood mononuclear cells.

The global demand for natural antioxidants, such as HT, is rising as consumer awareness of their health benefits increases and preferences shift toward natural alternatives to synthetic additives [23,24]. The market for HT is experiencing significant growth, which is propelled by continuous research into its health benefits, increasing applications across diverse industries, and improvements in extraction and sustainable sourcing technologies [25,26,27,28,29].

This review intends to investigate the recent comprehension of hydroxytyrosol’s molecular pathways in regulating inflammation and apoptosis. Using an extensive assessment of preclinical and clinical investigations, our objective is to provide an overview of hydroxytyrosol’s therapeutic abilities and its implications for the prevention and treatment of disorders associated with inflammation and apoptosis.

## 2. Properties, Bioavailability, and Extraction of Hydroxytyrosol

HT, also known as 3,4-dihydroxyphenylethanol, is a phenolic alcohol compound with a molecular weight of 154.16 g/mol and a chemical formula of C₈H₁₀O₃ that is extracted from the olive tree as a byproduct of olive oil [30]. This substance is found in the soluble fraction of minor components in extra virgin olive oil and is highly concentrated in the fruits and leaves of the olive tree (*Olea europaea* L.) (Figure 1) [31]. HT is the main compound found in olive oil. The surge in global interest in HT can be attributed to its wide-ranging health applications and biological effects [9,32,33]. These effects include cardioprotective, anticancer, neuroprotective, eye and skin protective, antibacterial, and desirable endocrine effects (Figure 1) [12,31,34,35]. Despite substantial research, the precise chemical pathways responsible for many of these activities have not yet been fully elucidated. The diverse range of HT biological actions was initially linked to its potent antioxidant potential. HT functions as an antioxidant by scavenging free radicals and chelating metal ions. The o-dihydroxyphenyl moiety is responsible for the remarkable antioxidant effectiveness of HT [36,37]. This compound demonstrates extensive health-promoting characteristics. Due to these discoveries, HT has been recommended as a nutraceutical for the prevention and treatment of multiple diseases [31].

The efficacy of HT depends on its bioavailability, which is defined as the rate of absorption, distribution, metabolism, and excretion within the body (Figure 2). In general, the bioavailability of HT is relatively high compared to that of other phenolic compounds [31].

Multiple factors can affect the bioavailability of HT. A critical factor is the manner of consumption. HT exists in both its free form and as a conjugate with other chemicals, such as oleuropein, in olive oil and olives [38]. The interaction of HT with other substances can influence its release and absorption in the gastrointestinal tract. Moreover, the presence of other dietary constituents, like fiber and fat, may interact with HT, affecting its absorption and metabolism [39].

Another factor is the gastrointestinal environment. The gastrointestinal environment plays a crucial role in the bioavailability of HT. The stomach’s pH and the presence of digestive enzymes can affect the release and absorption of HT [40]. Additionally, the intestinal microbiota has the ability to metabolize HT, which may influence its bioavailability and biological activity. HT enters the small intestine and is quickly absorbed. It can readily cross cell membranes due to its hydrophilic characteristics and very small molecular size. Following absorption, it quickly metabolizes to produce a variety of metabolites, mainly glucuronides and sulfates, which are eliminated in the urine [39,41].

Because HT may pass across the blood–brain barrier, it can efficiently enter cells and tissues, including the brain. Its capacity to exhibit neuroprotective effects is increased by this behavior. After consumption, the chemical is mostly eliminated by urine after a few hours. Because of its short half-life, the body needs to consume it frequently in order to sustain high levels [40].

HT is derived from the hydrolysis of oleuropein, which is found in olives. This process occurs naturally when the olives mature, as well as artificially during the storage and processing of table olives [42]. During the process of crushing olives to extract their oil, three distinct layers enriched with polyphenols are obtained: olive mill effluent (watery layer), pomace (solid residue), and olive oil. Each of these layers contains different amounts of HT and other valuable polyphenols [43].

During hydrolysis, oleuropein is broken down to produce HT as a simpler compound. Hydrolysis involves the addition of water molecules, which aids in breaking the chemical bonds in oleuropein. Enzymes that are naturally present in olives, especially during ripening and storage, catalyze this reaction. As a result, HT becomes available in a form that can be extracted and utilized [43,44].

HT exhibits amphipathic properties, which means that it can be dissolved in both water and fats. This enables it to exist in a variety of forms, such as acetate-bound HT, free HT, and as a component of more complicated compounds, such as verbascoside, oleacein, and oleuropein [31]. This versatility in structure and solubility increases the availability of HT across different layers in olive products, making it a valuable compound for health-related applications [45].

Although there is limited research available about the toxicity of HT, the existing information and data show that HT is generally well-tolerated [25,26,27,46]. According to one study, giving Wistar rats oral dosages of pure HT at 5, 50, and 500 mg/kg/day for 13 weeks did not cause any adverse effects, such as alterations in the size or shape of their organs, sickness, or death [47]. The use of HT (5 mg/day) or its derivative was approved by the European Food Safety Authority (EFSA) in 2011. This approval depends on HT’s potential benefits in lowering the risk of cardiovascular disease and insulin resistance/diabetes and protection against oxidative damage and inflammation [48]. However, more systematic and extensive human research is required to identify the optimum dosage needed to obtain health benefits while minimizing any potential toxicity or side effects.

## 3. Anti-Inflammatory Property of Hydroxytyrosol

Inflammation is the body’s natural defense mechanism against harmful stimuli, such as infections or injuries, aiming to eradicate threats and promote healing [49,50]. It involves two main stages: initiation, in which immune cells like leukocytes are activated to release inflammatory mediators, and resolution, which is a carefully controlled process that restores balance through specialized cells and molecules [51]. HT plays a key role as an anti-inflammatory agent by modulating these processes, helping to reduce excessive inflammation and support the resolution phase, which is crucial for preventing chronic inflammatory conditions [52,53].

According to Utami, et al. [54], HT is the most attractive molecule among all the phenolic components of olives for investigation and utilization in the pharmaceutical, nutraceutical, medical, cosmetic, business, and food industries. In any case of inflammation, there will be increasing levels of inflammatory cytokines, such as TNF-α, IL-6, IL-8, and IL-1β. Inflammation triggers the activation of nuclear factor κβ (NF-κβ), which, in turn, initiates the transcription of several cytokine genes [55]. Thus, the suppression of NF-κβ has emerged as a goal in regulating inflammatory cytokines. Guo, et al. [56] have demonstrated that HT suppresses the increased expression of NF-κB and p53 in HaCat cells. Lopez, et al. [57] conducted further research to examine the impact of HT on the regulators of NF-κβ, which are (IκBα) and (IKKβ) (Figure 3). HT inhibits the activity of IκBα and IKKβ generated by TNF-α in HUVEC. As a result, the NF-κβ signaling pathway will be deactivated, leading to the cessation of inflammatory cytokine production. This may be observed through the downregulation of the chemokine (CAC) motif ligand 2 (CCL_2_) and prostaglandin-endoperoxidase synthase 2 (PTGS2) [57].

When it comes to the inflammatory cytokines, HT also demonstrates a suppressive effect, leading to the downregulation of their expression and minimizing inflammation [58]. Calabriso et al. [18] demonstrated the inhibition of TNF-α by HT in their study using PMA-induced HUVEC. The inhibitory impact of HT on TNF-α was likewise demonstrated in THP-1 monocytes generated by LPS. The included studies also reported the inhibition of interleukins (ILs) by HT. Regarding IL-1β, HT has demonstrated the ability to inhibit its increased expression in PMA-induced HUVEC, TNF-α-induced THP-1 monocyte, and UVA-induced dermal fibroblast. Additional interleukins, such as IL-6 and IL-8, were also examined. HT demonstrates the inhibition of IL-6 in TNF-α-stimulated THP-1 monocytes, human Simpson–Golabi–Behmel Syndrome (SGBS) adipocytes and UVA-exposed dermal fibroblasts [59]. IL-8 was observed to be inhibited in UVA-induced dermal fibroblasts [58].

In addition, HT has been proven to decrease the excessive production of cell adhesion molecules, which are involved in inflammation and immunological responses [60]. Additionally, HT hinders the functioning of phospholipase A2, COX_2_, lipoxygenase, iNOS, and myeloperoxidase, which are enzymes that play a role in synthesizing inflammatory mediators [61]. In summary, HT exhibits a versatile strategy for diminishing inflammation by specifically addressing different pathways and agents involved in the inflammatory process. HT not only has anti-inflammatory actions but it also demonstrates antioxidative capabilities. HT has the ability to decrease the amount of ROS and free radicals, which contribute to oxidative stress and inflammation [57,62,63].

## 4. Antiapoptotic Properties of Hydroxytyrosol

Apoptosis is a critical process of regulated cell death that takes place not just in response to cell injury or external stress but also during normal development and morphogenesis [64]. Apoptosis is highly regulated by various groups of executioner and regulatory molecules [65,66]. The dysregulation of apoptosis results in pathologies such as cancer, abnormal development, and degenerative illnesses [67,68,69]. The extrinsic and intrinsic pathways are the two main mechanisms that contribute to the induction of apoptosis (Figure 4). The intrinsic pathway is a mitochondrial-mediated pathway that is activated by intracellular signals, whereas the extrinsic pathway refers to a death receptor-mediated pathway that is induced by extracellular signals [70]. The extrinsic and intrinsic apoptotic routes both lead to the same endpoint, which is the execution pathway [65,71,72]. The tumor necrosis factor (TNF) receptor superfamily, such as Fas and TNF-related apoptosis-inducing ligand (TRAIL) receptors, TNF receptors (TNFR), and Fas (CD95), plays a crucial role in the extrinsic pathway [73,74]. Aside from the external stimuli, internal inputs, such as replication stress, oxidative stress, growth factor deprivation, and DNA damage, could also regulate or recruit apoptosis [75]. Members of the Bcl family, in particular Bax and Bcl-2, are crucial mediators for the intrinsic apoptotic process [71,76]. Both intrinsic and extrinsic pathways activate caspase 3, which is responsible for nuclear apoptosis through the activation of poly (ADP-ribose) polymerase (PARP) [71,77].

Research into the antiapoptotic properties of HT is currently in its primary phase and requires further work to broadly understand the underlying mechanisms. However, this property of HT is related to the antioxidant and anti-inflammatory properties and HT’s ability to modulate cell signaling pathways. Abd Elmaksoud et al. [15] have explained in their experimental study that HT functions as an antiapoptotic molecule by efficiently reducing levels of colon malondialdehyde (MDA), nitric oxide (NO), and myeloperoxidase (MPO) while significantly elevating levels of superoxide dismutase (SOD), glutathione peroxidase (GPX), and catalase. Additionally, it induces the downregulation of pro-inflammatory cytokines, such as IL-1β, TNF-α, IL10, Cox_2_, iNOs, TGF-β, monocyte chemoattractant protein-1 (MPC1), and NF-kB. In addition, the expression of the apoptotic gene Bax was decreased, and the expression of the antiapoptotic gene Bcl2 was increased following treatment with HT when compared to the untreated ulcerative colitis group [15]. Figure 5 illustrates a simple explanation of how HT can exert an antiapoptotic effect on the apoptotic pathway.

Facchini et al. [78], in their study on chondrocyte cells, proved that HT reduced the expression of mRNAs for pro-inflammatory cytokines, thereby antagonizing the activation of pro-inflammatory pathways, such as the NF-κB pathway. This was supported by a previous study conducted by Zhang et al. [79], who stated that the inhibition of NF-κB by HT downregulated pro-inflammatory and catabolic genes, such as those coding for iNOS, COX_2_, and MMPs, and prevented cell damage and apoptosis. Another study conducted by Burattini et al. [17] demonstrated that HT and HT laurate (5–40 µM) have protective effects against H_2_O_2_-induced apoptotic death in different cell lines by altering the mitochondrial or DNA pathways. This study also explored whether HT could prevent apoptosis through the inhibition of autophagic vacuoles, which is a cell-protective mechanism.

Wang et al. [80] discovered that when C2C12 myotubes were treated with HT-acetate at concentrations ranging from 1 to 50 µM for a duration of 24 h, it effectively reduced the damage to mitochondria caused by tert-butylhydroperoxide (t-BHP). Additionally, it prevented the cleavage of optic atrophy 1 (OPA-1) and muscle degradation. Furthermore, this treatment increased the capacity for oxygen consumption, ATP production, and the activities of mitochondrial complexes I, II, and V. It also enhanced the expression of myosin heavy chain. In addition, the HT treatment significantly enhanced cell viability.

Furthermore, HT demonstrated its ability to provide protection against oxidative stress, inflammation, hyperglycemia, and hyperlipidemia in animal models by decreasing levels of inflammatory cytokines, serum CRP, and TBARS in the liver, kidney, heart, and pancreas while increasing mitochondrial biogenesis, oxygen consumption capacity, ATP production, and cell viability. All of this indicated that HT has antiapoptotic effects on different types of cells [80,81,82,83,84].

## 5. Preclinical Studies: Evidence Supporting the Anti-Inflammatory and Antiapoptotic Effects of Hydroxytyrosol

This section presents a summary of the preclinical research that supports the therapeutic potential of HT in reducing inflammation and apoptosis. It emphasizes the probable uses of HT in preventing and treating different disorders. Table 1 summarizes the studies conducted in the past 5 years that showed the ability of HT to reduce the inflammatory process and apoptosis in various diseases using in vitro or in vivo models. The findings demonstrated that HT affects many diseases and can mitigate inflammation, as well as apoptosis, through different mechanisms of action and signaling pathways, mainly by reducing the pro-inflammatory cytokines and downregulating the apoptotic genes.

## 6. Discussion and Future Perspective

Olive oil is widely recognized for its multiplicity of health benefits, which are mainly due to its substantial concentration of monounsaturated fats, antioxidants, and other bioactive compounds, especially in Mediterranean countries [94]. HT is a crucial compound extracted from the phenolic portion of olive oil. It is known for its ability to promote health, mostly because of its strong antioxidant properties and anti-inflammatory effects. Previous research has shown that HT has beneficial effects in preventing oxidative stress, inflammation, hyperglycemia, and hyperlipidemia [32].

HT is particularly noteworthy for its capacity to regulate the inflammatory response. Chronic inflammation has a crucial role in the development of various diseases, such as cardiovascular problems, neurological diseases, and malignancies. HT has demonstrated the ability to inhibit the generation of pro-inflammatory cytokines, including TNF-α, IL-1β, and IL-6, by reducing the activation of NF-κB and other inflammation-related signaling pathways. The anti-inflammatory impact of this substance is reinforced by its ability to decrease oxidative stress, which is closely connected to inflammatory reactions. Hydroxytyrosol’s ability to simultaneously reduce oxidative stress and block inflammatory mediators makes it a potent chemical for preventing and treating disorders associated with inflammation [89,90].

HT not only has anti-inflammatory qualities but also has notable antiapoptotic effects. These activities are important for safeguarding cells against programmed cell death in many clinical circumstances. Apoptosis, although an essential physiological process, can cause tissue damage in instances such as ischemia injury, dementia, and some autoimmune illnesses when it is not well regulated. HT has shown the capacity to regulate important processes involved in cell death, such as the intrinsic and extrinsic pathways. It does this by affecting the expression of Bcl-2 family proteins and blocking caspase activation. These methods aid in maintaining the structural integrity of cells during stressful settings, emphasizing the potential of HT in therapeutic approaches targeting the reduction of undesired apoptosis [14,19,95].

HT has the potential to be used therapeutically to reduce inflammation and apoptosis, which could lead to the development of new treatments for various disorders. Nevertheless, in order to apply these preclinical discoveries to practical medical treatment, it is essential to possess an extensive understanding of the pharmacokinetics, bioavailability, and long-term safety of HT in human beings [33]. Although existing studies provide a solid basis, it is imperative for future research to prioritize conducting clinical trials in order to substantiate the effectiveness of HT in different illness scenarios. In addition, investigating the synergistic effects of this molecule with other bioactive substances could increase its therapeutic efficacy.

## 7. Conclusions

In summary, HT displays significant potential as a natural agent for mitigating inflammation and preventing cellular damage (apoptosis). Its potent antioxidant effects enable it to protect cells and mitigate damaging effects within the body. This natural chemical, which is derived from olives, could potentially be valuable in treating many health problems associated with inflammation and cellular apoptosis, thereby providing a safer, plant-based alternative in therapeutic applications. Additional research will further clarify its benefits and broaden its potential use in healthcare.

## Figures and Tables

**Figure 1 pharmaceutics-16-01504-f001:**
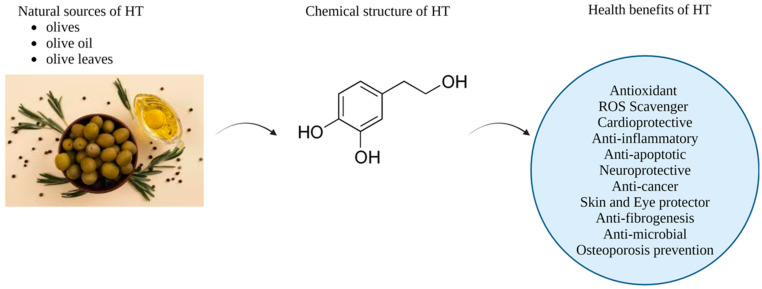
Natural sources, chemical structure, and health benefits of HT. Created with BioRender.com. (https://www.biorender.com) (accessed on 26 August 2024).

**Figure 2 pharmaceutics-16-01504-f002:**
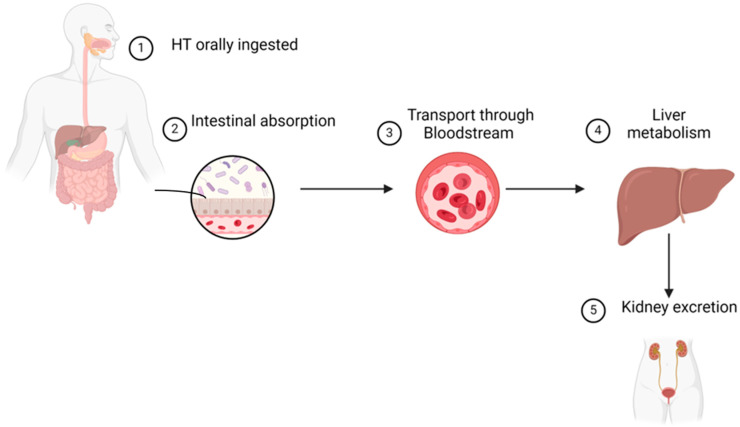
Bioavailability of HT inside the human body. Created with BioRender.com. https://app.biorender.com (accessed on 2 October 2024).

**Figure 3 pharmaceutics-16-01504-f003:**
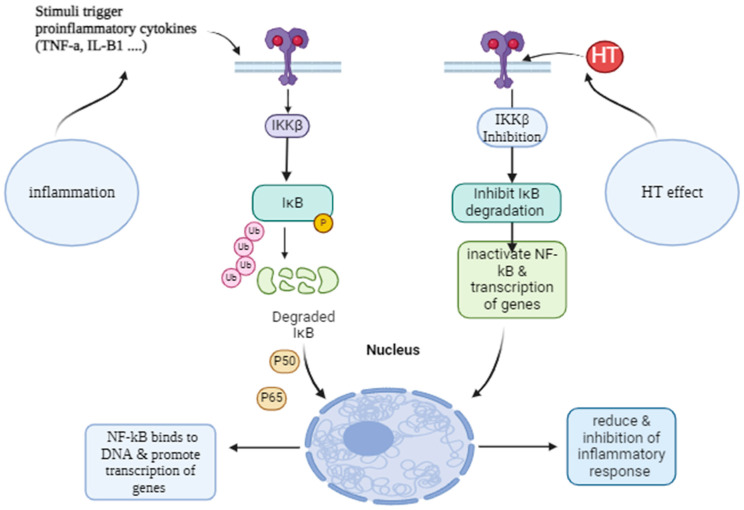
Anti-inflammatory effect of HT on NF-kB signaling pathway. Created with BioRender.com. https://app.biorender.com (accessed on 28 August 2024).

**Figure 4 pharmaceutics-16-01504-f004:**
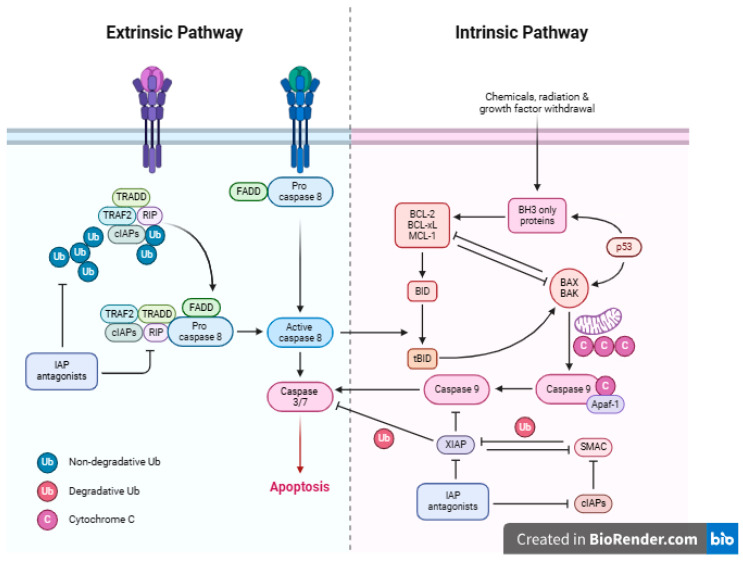
Extrinsic and intrinsic apoptosis pathway. Created with BioRender.com. (https://app.biorender.com (accessed on 28 August 2024).

**Figure 5 pharmaceutics-16-01504-f005:**
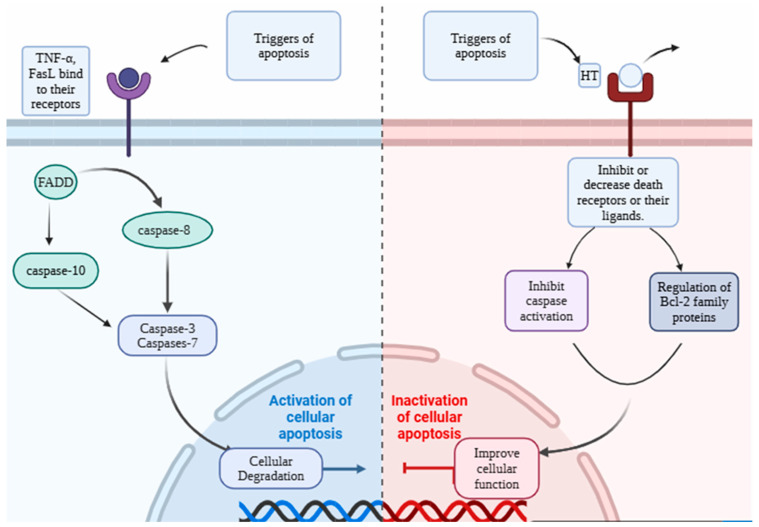
Antiapoptotic effect of HT on the extrinsic apoptotic pathway. Created with BioRender.com. https://app.biorender.com (accessed on 30 August 2024).

**Table 1 pharmaceutics-16-01504-t001:** Studies conducted in the past 5 years that showed the ability of HT to reduce the inflammatory process and apoptosis in various diseases using in vitro or in vivo models.

Author	Model/Cell Type	Dose of HT	Route of Treatment	Findings	Reference
(Yonezawa et al., 2019)	5-week-old male Sprague–Dawley rat	500 mg/kg	Orally	- Inhibits rat paw swelling. - Exerts sustained anti-inflammatory effects. - Decreases PGE_2_ production. - Prevents gastric damage.	[85]
(Abd Elmaksoud et al., 2021)	6- to 8-week-old male albino rats	150 mg/kg	Orally	- Reduces the mortality rate and disease activity index. - Reduces oxidative stress and inflammation. - Elevates SOD, CAT, and GPX levels with upregulation of antiapoptotic gene Bcl_2_. - Downregulates the pro-inflammatory cytokines and apoptotic gene Bax.	[15]
(Alblihed 2021)	8-week-old male Swiss mice	20 and 40 mg/kg	Orally	- Increases the survival rate. - Decreases the LDH level, inflammatory proteins, and inflammatory cytokines. - Decreases the mRNA expression of iNOS and NO production. - Downregulates p38MAPK expression and decreases NF-κB activity. - Restores the depleted antioxidants, elevated SOD activity, and decreased the elevated MDA level.	[16]
(Chen et al., 2021)	HK-2 cell line.	6.25, 12.5, 25, 50, 100 μM	Supplemented with culture media	- Protects against cytotoxicity. - Reduces the elevation of mRNA levels and protein levels of CKLF1. - Reduce the over-expression of CKLF1 and NF-kB. - Alleviate oxidative stress. - Inhibit cell apoptosis.	[20]
	Male C57BL/6 mice.	20, 40, 80 mg/kg	Intragastric	- Reverses the weight loss. - Reduces the elevated serum creatinine and BUN levels. - Improves pathological damage. - Suppressing the expression of TNF-α and IL-1β. - Decreases expression of CKLF1 and NF-kB. - Resists oxidative stress and inhibits apoptosis. - Exerted protective effects on nephrotoxicity.	
(Li et al., 2024)	SH-SY5Y, BV2, and HT22 cell lines.	10, 20, 40 and 80 μmol/L	Supplemented to culture media	- Enhances cell viability. - Suppressed NO levels. - Protects mitochondrial function, enhances network connectivity, and prohibits the compromise of mitochondrial membrane integrity. - Inhibits IL-1β via affecting the expression of BDNF/TrkB/CREB protein.	[86]
	6- to 8-week-old male Sprague–Dawley rats and 6- to 8-week-old male ICR mice	50, 150, 450 mg/kg	Intragastric	- Enhances sucrose preference. - Diminished the elevated immobility times. - Elevates neurotransmitters and improves behavioral outcomes. - Alleviates the depressive phenotype. - Upregulates the BDNF/TrkB levels and phosphorylation of the transcription factor CREB. - Reverses the elevation in inflammatory cytokines. - Elevates tryptophan levels and suppresses Kyn/Trp-QUIN metabolic pathway.	
(Yu et al., 2022)	HNPCs	20, 50, and 100 μM	Supplemented with culture media	- Inhibits the secretion of inflammatory cytokines. - Decreases the mRNA level of NF-kB and the nuclear translocation rate of p65. - Reduces the increase in endogenous ROS levels. - Reduces the number of abnormal mitochondria. - Decreases the production of proapoptotic indicators c-caspase3 and Bax and increases the secretion of antiapoptotic indicators Bcl-2.	[14]
	2-month-old Sprague–Dawley rats	10 μL	Intrathecal injection	- Reduces MMP-13 expression and prevents the loss of col-2 expression. - Alleviates proteoglycans loss. - Inhibits the expression of IL-1β, IL-6, COX_2_, and iNOS. - Alleviates neuropathic pain. - Inhibits PI3K/AKT and ERK signaling pathways.	
(Yao et al., 2019)	HUVECs	12.5, 25, 50, 100 and 200 µmol/L	Supplemented with culture media	- Decreases the mRNA expressions of IL-1β, IL-6, and CCL2. - Inhibits oxidative stress. - Downregulates the expressions of TNFRSF1A protein and mRNA. - Upregulates SIRT6 expression. - Inhibits protein expression and nuclear translocation of PKM2.	[87]
	Endothelium-specific Sirt6 knockout (Sirt6^endo−/−)^ mice	5, 10, 20 mg/kg	Gavage dosing	- Decreases the concentrations of TNF and IL-1β. - Decreases the mRNA expressions of TNF, IL-1β, IL-6, and CCL2. - Increases SIRT6 protein and mRNA expression. - Decreases PKM2 protein expression.	
(Zhao et al., 2021)	7- to 8-week-old male C57BL/6 mice	50, 100, 200 mg/kg	Gastric gavage	- Increases sugar water preference. - Slows immobility time. - Inhibits ROS production. - Increases the SOD activity and decreases MDA levels. - Decreases the number of microglial cells. - Decreases the levels of IL-1β and TNF-α. - Elevates the number of GFAP-positive cells and enhances GFAP expression. - Increases BDNF, p-TrkB, and p-CREB signaling pathways.	[88]
(Chen et al., 2019)	DPCs were isolated from 5-week-old rat vibrissae	75 μM	Supplemented with culture media	- Induces LC3 expression. - Inhibits ROS production. - Reduces the expression of caspase-3 and the number of apoptotic markers. - Decreases the apoptotic patterns. - Diminishes the release of inflammatory cytokines. - Upregulates the expression of growth factors.	[19]
(Fki et al., 2020)	10-week-old male Swiss rats	16 mg/kg	Orally	- Decreases the plasma lipid content. - Lowers plasma levels of glucose, insulin, HOMA-IR, AST, and ALT. - Decreases oxidative stress. - Shows depletion of plasma leptin and nucleus death. - Decreases the expression of P53, COX_2_, and TNF-α. - Improves Bcl-2 protein expression.	
(Yu et al., 2020)	Macrophage RAW 264.7	50 and 100 μM	Supplemented with culture media	- Reduces COX_2_ level, RNA levels, and expression of TNF-α, IL-1β, and IL-6. - Increased the levels of anti-inflammatory cytokines, such as IL-10 and IL-4. - Decreased p-ERK1/2.	[89]
	6- to 8-week-old male C57BL/6 mice	100 mg/ kg,	Orally	- Inhibits CD11c and COX_2_ levels. - Reduces the RNA levels of TNF-α, IL-1β, and IL-6. - Increases IL-10 and IL-4. - Elevates CD206 expression in liver tissues. - Restrained the elevation of ALT, AST, and total bilirubin level.	
(Chen et al., 2023)	HaCaT cell line	25, 50, 100, and 200 µM	Supplemented with culture media	- Suppresses the production of IL-8. - Decreases the expression level of TNF-a and the production of IL-6. - Suppresses the levels of h-BD2, S100A7, S100A8, and S100A9. - Increases the percentage of cells in the G1 phase with a concomitant decrease in cells in the S phase.	[90]
(Zhang et al., 2021)	RAW 264.7 cells. MC3T3-E1 cells.	1, 2.5, 5, 10, or 20 μM. 5, 10, 20, or 50 μM	Supplemented with culture media	- Inhibits the expression of osteoclast marker genes. - Attenuates the mtROS and MMP levels. - Reduces the phosphorylation of ERK and JNK signaling pathways. - Decreases H_2_O_2_-induced apoptosis. - Recovers ALP activity. - Increases the mRNA expression of osteogenic genes. - Attenuates mtROS levels and the loss of MMP.	[91]
	6-week-old male C57BL/6 mice	20 mg/kg	Intragastric	- Prevents extensive alveolar bone loss. - Suppresses bone destruction - Increases the level of RUNX2. - Decreases the level of 8-OHdG to a much lower level.	
Sirangelo et al., 2022)	Embryonic rat cardiac tissue-derived H9c2 cardiomyoblasts	20, 50, and 70 µM	Supplemented with culture media	- Reduces Dox-induced toxicity. - Increases the antiapoptotic Bcl-2 protein expression. - Decreases pro-apoptotic Bax protein expression. - Suppresses caspase 3 activation. - Reduces the level of γ-H2AX.	[92]
(Yao et al., 2021)	8-week-old male ApoE−/− mice	20 mg/kg	Gavage dosing	- Decreases the deposition of lipid. - Decreases plaque formation. - Suppresses caspase-1 activation, GSDMD, and GSDMD-N. - Decreases level of TNF-α and IL-1β. - Decreases mRNA expressions of HDAC11 protein.	[93]
	HUVECs cell line (CRL-1730 cells)	25, 50, and 100 μmol/L	Supplemented with culture media	- Decreases the protein expressions and activity caspase-1. - Reduces the protein expressions of GSDMD and GSDMD-N. - Diminishes the release of pro-inflammatory IL-1β and IL-6. - Decreases mRNA expressions of HDAC11 protein.	

## Data Availability

Not applicable.

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
