# Peer review of "Hydroxytyrosol: A Promising Therapeutic Agent for Mitigating Inflammation and Apoptosis"

_pharmaceutics, 2024, doi:10.3390/pharmaceutics16121504_

Round 1
Reviewer 1 Report
Comments and Suggestions for Authors
The article, authored by Wafa Ali Batarfi et al., provides a review of the potential therapeutic effects of hydroxytyrosol (HT), a phenolic compound found in olive oil, particularly in managing inflammation and apoptosis-related disorders.
The review is well structured, I liked the way the mechanisms were explained and the fact that suggestive images were used for the content.
I would only comment on the toxicity part. The authors state that HT is well tolerated, but cite only one article. I would suggest to add more references in this part as I found several articles on the same topic, more recent than the cited article, even an article dedicated to this issue, requested by EFSA: Safety of hydroxytyrosol as a novel food pursuant to Regulation (EC) No 258/97
Reviewer 2 Report
Comments and Suggestions for Authors
Comments for pharmaceutics-3288385-peer-review-v1
1. Please check the biological names are written in italics or follow the accepted guidelines for writing the scientific names.
2. The heading 2. Hydroxytyrosol should come with some headings Antioxidant potential of Hydroxytyrosol.
3. Line no 69 to 89 talks about benefits and then 93-103 explain about toxicity of HT and then from 104 again benefits. I think the paragraph are not arranged sequentially. The toxicity of HT topics can be dealt separately at the last or some other paragraph.
4. The last paragraph 128-137 of second paragraph HT should come under Introduction line 40 recent findings…..
5. The market of HT line 108-120 should also be in introduction.
6. The third paragraph should be the anti-inflammatory properties of HT then anti-apoptotic property of HT and then preclinical studies….
7. Paragraph 121-126 is also not clear. The paragraph need to be restructured.
8. The representation of Cox2, H2O2-induced and other abbreviation needs to be checked and may be added at the end of the article.
9. The discussion on the mechanisms of inflammation and apoptosis can be found in many of the article and needs to be removed. The article is less about HT rather more about the mechanism of apoptosis and inflammation.
10. The authors want to explain the antidiabetics effects of HT in the last paragraph of anti-apoptosis effect of HT or to explain inflammation and apoptosis.
11. There is many studies which is missing in this article about the apoptosis effect of HT in cancer cells. The authors discusses the anti-apoptotic effects of HT and explain its importance in Discussion and future perspectives whereas need to be discussed in start of the Anti-apoptotic effects of HT.
12. Conclusion is deviating and do not correspond to the topics discussed in this review.
13. Tables needs to be more simplified especially the findings. The table most of the studies are related to the inflammation and less of antiapoptotic activity.
Round 2
Reviewer 2 Report
Comments and Suggestions for Authors
Comments for pharmaceutics-3288385-peer-review-v2
1. Still the introduction is deviating from the main topics. It should be more concise and related to the title.
2. The section 2 has been improved.
3. The section 3 and section 4 is too broad and mostly discussed about apoptosis and anti-inflammatory mechanism. I had given the comments in the first review. The authors defended that they keep it for the coverage and understanding for the vast readers.
